# OmniV2V: Versatile Video Generation and Editing via Dynamic Content Manipulation

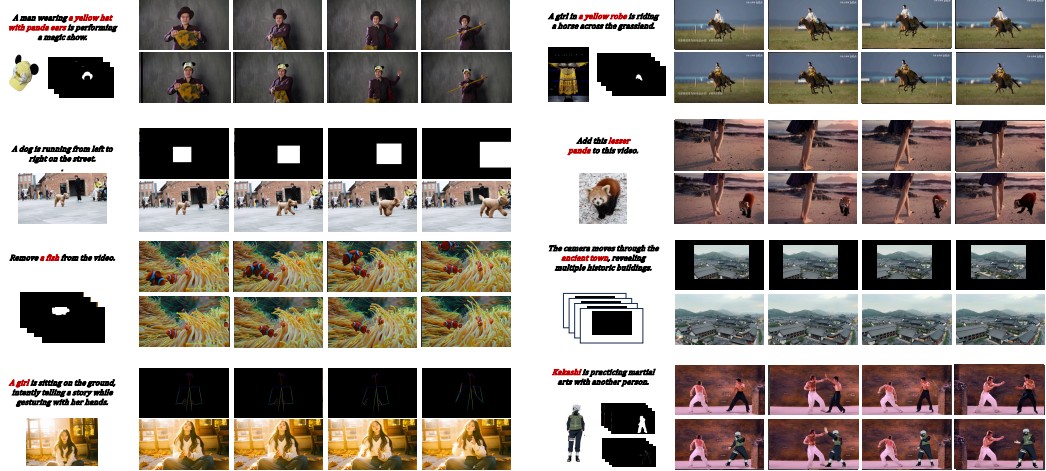

Figure 1: **OmniV2V comprehensive capability demonstration**. We showcase the excellent generation and editing results of OmniV2V, with the original input and the generated videos for each task displayed in the figure.

## Abstract

The emergence of Diffusion Transformers (DiT) has brought significant advancements to video generation, especially in text-to-video and image-to-video tasks. Although video generation is widely applied in various fields, most existing models are limited to single scenarios and cannot perform diverse video generation and editing through dynamic content manipulation. We propose OmniV2V, a video model capable of generating and editing videos across different scenarios based on various operations, including: object movement, object addition, mask-guided video edit, try-on, inpainting, outpainting, human animation, and controllable character video synthesis. We explore a unified dynamic content manipulation injection module, which effectively integrates the requirements of the above tasks. In addition, we design a visual-text instruction module based on LLaVA, enabling the model to effectively understand the correspondence between visual content and instructions. Furthermore, we build a comprehensive multi-task data processing system. Since there is data overlap among various tasks, this system can efficiently provide data augmentation. Using this system, we construct a multi-type, multi-scenario OmniV2V dataset and its corresponding OmniV2V-Test benchmark. Extensive experiments show that OmniV2V works as well as, and sometimes better than, the best existing open-source and commercial models for many video generation and editing tasks. The source code will be released publicly.

## 1 Introduction

In recent years, Diffusion Transformers (DiT) has led to significant advancements in video generation models. Text-to-video and image-to-video generation Zhou et al. (2024); Blattmann et al. (2023a;b);

Guo et al. (2023); Zhou et al. (2022); Gupta et al. (2023); Wang et al. (2023); Ho et al. (2022); Brooks et al. (2022); Wang et al. (2020); Singer et al. (2022); Li et al. (2018); Villegas et al. (2022); Lin et al. (2025a) have attracted increasing attention as they approach the threshold of practical application. In addition, downstream tasks based on these pre-trained models have become increasingly diverse, such as object movement, object addition, mask-guided video edit, try-on, human animation, controllable character video synthesis, inpainting, and outpainting. These tasks involve different content inputs, reflecting the dynamic and complex nature of video generation and editing.

Current video generation models perform well on specific tasks, but each new task typically requires dedicated modules and fine-tuning. For example, in character image animation, methods like Animate Anyone Hu (2024) use ReferenceNet Hu (2024) to fit the reference character, while pose is driven by adding it to the noise. For object addition, Get in Video Zhuang et al. (2025) uses a T5 encoder Chung et al. (2024) to input instructions and compresses the original video and reference image through a 3D VAE. In try-on, methods such as Tunnel Try-on Xu et al. (2024) and Stableviton Kim et al. (2024) use ReferenceNet or ControlNet Zhang et al. (2023a) to inject clothing information, performing clothing replacement by concatenating the source video, mask video, and other information along the channel dimension. Although these approaches achieve impressive results, their complex structures and lack of interoperability lead to significant waste of computational and data resources. We observe that leveraging commonalities among tasks can help models better understand and perform across tasks. For example, in mask-guided video editing, the role of text is often overlooked, either the text branch is removed or only captions are encoded, largely ignoring the relationship between text and image. In object movement task, boundingbox information dominates, and textual information is neglected.

To address the high deployment and training costs associated with task-specific video generation models, we propose **OmniV2V, a unified framework capable of both video generation and editing** according to diverse user operations. Building on the mainstream MM-DiT architecture, we adopt HunyuanVideo Kong et al. (2024b) as our backbone to maximize model capacity and performance. To enable flexible and effective handling of various tasks, we first introduce a unified **dynamic content manipulation injection module**. This module integrates all dynamic content operation inputs such as reference images, background videos, pose videos, and mask videos into a single framework, leveraging multi-modal information. To distinguish between different visual modalities across tasks, we employ a dynamic routing strategy that adaptively adjusts model inputs, allowing the model to discern which content should be preserved and which should be modified. Furthermore, we design a **visual-text instruction module** based on LLaVA Liu et al. (2023), enabling the model to better understand and align visual content with textual instructions. Unlike HunyuanVideo, which only uses LLaVA for text understanding and does not establish explicit connections between text and visual content, our approach ensures that the model can accurately associate reference images with textual concepts in the caption. This alignment is crucial for tasks involving reference images, as it allows the reference character to act according to the given instructions.

To construct comprehensive datasets for various tasks, we utilize a multi-task data processing system and leverage various open-source tools in combination to efficiently filter and select high-quality data. To comprehensively evaluate our model's performance on different tasks, we build task-specific benchmarks. By comparing with existing open-source and commercial methods, it demonstrates the strong competitiveness of our model. Extensive experiments show that our design can effectively unify various video generation and editing tasks, significantly improving video dynamics and reference consistency. In summary, our contributions can be summarized as follows:

- We propose **OmniV2V**, a unified video generation and editing framework that supports a wide range of user operations, including object addition and replacement, video inpainting and outpainting, pose-guided generation, and more.

- We introduce a **unified dynamic content manipulation injection module** that flexibly integrates multi-modal inputs (e.g., reference images, background videos, and pose videos) and employs a dynamic routing strategy to distinguish and process different visual modalities.

- We design a **visual-text instruction module** based on LLaVA, enabling the model to effectively align and understand the correspondence between visual content and textual instructions for more accurate and controllable video editing.

- We construct **comprehensive multi-task datasets** and **task-specific benchmarks** using a multi-task data processing system and open-source tools, facilitating robust evaluation and demonstrating the competitiveness of our approach against existing methods.

## 2 RELATED WORK

Recent advancements in video generation have been significantly propelled by diffusion models, which have evolved from static image synthesis (Rombach et al., 2022; Li et al., 2024; Labs, 2024) to video generation (Hong et al., 2022; Zhang et al., 2023c). The field has seen substantial progress with the development of large-scale frameworks (Liu et al., 2024; Yang et al., 2024; Kong et al., 2024a; Wang et al., 2025; Zhou et al., 2024), which demonstrate unprecedented high-quality content creation and a diverse array of generated results through extensive training on video-text pairs.

However, existing methods primarily focus on either text-guided video generation (Lin et al., 2025b) or video generation based on a single reference image (Gao et al., 2023; Xu et al., 2025). These approaches often struggle to provide fine-grained control over the generated content and precise concept-driven editing, a limitation that persists despite advancements in multi-condition control. While pioneering work such as VACE (Jiang et al., 2025) enables multi-condition capabilities through multi-modal modeling, it fails to maintain identity consistency due to the excessive number of training tasks. In this study, we focus on video editing and aim to enhance the consistency of characters or objects through sophisticated data processing and the design of a video injection model.

## 3 METHODS

We propose a unified video editing approach, **OmniV2V**, which supports various primary control signals as input to generate corresponding videos using textual information. Specifically, our method allows for image, video, mask video, and pose video as conditional inputs to produce video content specified by text. This enables key video editing tasks such as object replacement, object addition, instruction-based editing, video inpainting, outpainting, pose-driven editing, and video face swapping. In detail, we introduce a unified dynamic content manipulation injection module that categorizes conditional inputs into image signals, mask signals, and pose signals, achieving decoupled processing and conditional fusion of these three types. Through a dynamic conditional training strategy, the model is capable of understanding individual signals while also integrating multiple signals, thereby enhancing the control capability of each signal through multi-signal comprehension. Additionally, we propose an instruction-based editing method based on LLaVA, which leverages a multimodal understanding model to effectively interpret human instructions while integrating image signal comprehension, thus enabling the conditional injection from text-image signals to video generation.

### 3.1 UNIFIED DYNAMIC CONTENT MANIPULATION INJECTION

Taking the controllable character video synthesis task as an example, we first resize the reference images $I_1$ and $I_2$ to match the dimensions of the target video. We then use the 3DVAE pretrained by HunyuanVideo13B to map the reference images $I_1$ and $I_2$ from the image space to the latent space, obtaining latent representations $v_1$ and $v_2$, where $w$ and $h$ denote the width and height of the latent, and $c$ is the feature dimension. These latents are then processed by Tokenizer1 to obtain $t_1$ and $t_2$. Similarly, the noise video, masked video, mask video, and pose video are passed through the 3DVAE to obtain $v_{\text{noise}}$, $v_{\text{md}}$, $v_{\text{mv}}$, and $v_p$, respectively. Next, $v_{\text{noise}}$ are processed by Tokenizer1 ($K_1$) to obtain and $T_{\text{noise}}$. The pose feature $v_p$ is processed by PoseNet and Tokenizer3 ($K_3$, initialized with the weights of Tokenizer1) to obtain $T_p$.

$$T_{\text{noise}} = K_1(v_{\text{noise}}), \quad T_p = K_3(\text{PoseNet}(v_p)) \tag{1}$$

**Latent-Fusion Video Tokenizer.** Since the masked source video and mask video contain overlapping information, where the masks in the mask video correspond to the masked regions in the source video. We employ a latent fusion tokenizer to merge these two streams of tokens into a single set, thereby effectively compressing the conditional information. Concretely, the 3D-VAE encoder encodes both videos from the RGB channel into 16-dimensional latent representations. These features are then concatenated to form a 32-dimensional feature vector. The original tokenizer in the pretrained HunyuanVideo model consists of a 3D convolutional network that maps the 16-dimensional features into a sequence of tokens. To leverage the robust tokenization capabilities of the pretrained tokenizer, our latent fusion tokenizer inherits its weights and pads zeros in the 3D convolutional layer to accommodate the 32-dimensional input. This process yields a set of fused tokens $T_M$ that integrate information from both the source and mask videos.

$$T_M = K_2(\text{ChannelCat}(v_{\text{md}}, v_{\text{mv}})), \tag{2}$$

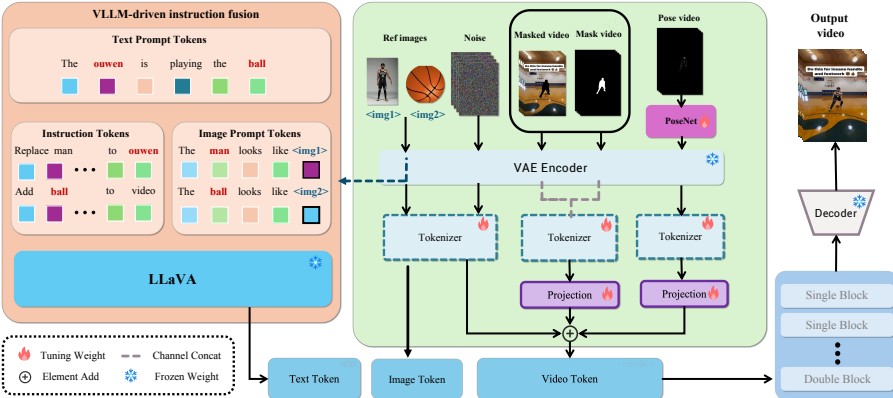

Figure 2: The framework of OmniV2V. It consists of two main modules: a unified information injection module for integrating task requirements and a visual-text instruction module for understanding visual-instruction correspondence.

**Token Fusion.** After obtaining the feature-aligned video tokens $T_M$ from the masked source and mask videos, the pose tokens $T_P$, and the noisy latent tokens $T_{noise}$, a key challenge is to establish effective correlations among them so that the noisy latent tokens can adequately incorporate the conditional information. Previous image-editing approaches often utilize adapter-based methods to inject conditions into the latent space. However, for the MMDIT-based HunyuanVideo model, the large parameter count, high-dimensional feature space, and long video token sequences make it difficult for a newly introduced module to map the conditional features into the latent space efficiently. To address this, we propose a more efficient video condition injection mechanism that merges all tokens. Specifically, we first apply a fully connected (FC) layer **Projection** to the two sets of condition tokens, mapping them into the latent space of the video tokens. With the aligned video, mask, and pose tokens, we directly sum them to form a new set of tokens. This approach enables effective injection of conditional information into the video tokens. Furthermore, the learnable Projection preceding the token addition allows the model to selectively retain or discard features, ensuring that only the most salient conditional information is incorporated.

We then sum $T_{noise}$, $T_p$, and $T_M$, and concatenate the result with $t_1$ and $t_2$ along the token dimension, together with $T_R$, to obtain the final input $H$, as shown in the following equation:

$$H = \text{TokenCat}\left(t_1, t_2, \{Projection(T_M) + T_{noise} + Projection(T_p)\}\right) \quad (3)$$

**PoseNet.** Effectively inputting pose-guided information into the model poses a challenge. Since our model is built on the HunyuanVideo framework, a video generation architecture based on MM-DiT. We considered two commonly used conditional injection strategies from MM-DiT: (1)Token Addition (as shown in figure 3(a)): This involves encoding the pose video into pose tokens using a tokenizer and then adding them element-wise to the video tokens. (2) ControlNet-based Method (as shown in figure 3(b)): This involves extracting pose information through an additional adapter network and injecting it between the layers of the HunyuanVideo model. However, both methods were initially designed for image generation and exhibit significant limitations when applied to video generation tasks. In the Token Add approach, we found that pose information tends to leave residual artifacts in the generated video, a problem that requires extended training time to mitigate. As for the ControlNet method, since pose video inherently contains relatively sparse information and ControlNet's structure is complex with a large number of parameters, the model struggles to effectively learn the crucial pose-guided signals, thereby affecting the injection effectiveness.

**Dynamic Content Manipulation Injection.** There are multiple video conditions in the model, but it is not always necessary to use all of them during editing. For example, in some cases, the pose video or mask video alone may suffice to generate the output, while in others, the mask video may need to be combined with the source video. To facilitate flexible video editing with arbitrary combinations of input conditions, we propose a dynamic content manipulation injection strategy. During training, we randomly set some of the conditional inputs to empty, enabling the model to learn to handle various combinations of conditional information. This unified training approach not only enhances the model's ability to process different sets of conditions but also improves its performance when editing based on a single condition, thereby significantly boosting its overall editing capabilities.

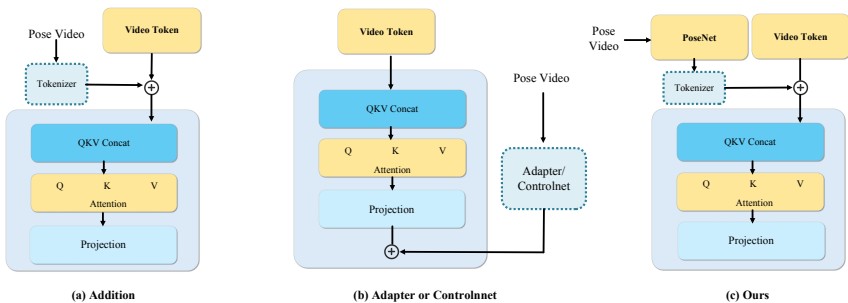

Figure 3: Three types of strategies for injecting pose information.

## 3.2 VLLM-DRIVEN INSTRUCTION-BASED EDITING

In previous video editing methods, the model typically overlooks the text prompt, resulting in output videos determined solely by the input masked video and mask video. However, the absence of a text prompt significantly limits the controllability of these methods. To address this limitation, we propose a VLLM-driven, instruction-based editing module that leverages the strong multimodal understanding capabilities of the pretrained LLaVA model to enable instruction-guided editing. Specifically, we decompose the text tokens in LLaVA into three components: (1) *instruction tokens*, which encode the user-provided editing instruction (i.e., what to edit); (2) *text prompt tokens*, which describe the content of the video the user wishes to generate; and (3) *image prompt tokens*, which incorporate a target image into the LLaVA text space. For example, to add a cat to a source video depicting a beach scene, the instruction prompt could be "Add a cat to the video," the text prompt might be "A cat is playing on the beach," and the image prompt could be "The cat looks like <image>." To clearly separate these three sets of tokens, we follow HunyuanCustom Hu et al. (2025) and insert a <SEP> token between them. The concatenated tokens are then input into the LLaVA model, which, through its autoregressive multimodal modeling capability, establishes correlations among the three sets of tokens to produce output text tokens.

Since the CLIP image encoder in LLaVA primarily captures high-level semantic features and may lose fine-grained image details, we additionally employ a 3D-VAE to encode the image, mapping it into the latent space while preserving detailed information. To effectively inject these image tokens into the model, we position the image along the temporal axis of the video tokens, specifically placing it before the first frame of the video tokens. Given that the pretrained video model possesses strong temporal modeling capabilities, the information from the image can be efficiently integrated into the video tokens via temporal modeling. In particular, the base video generation model (HunyuanVideo) utilizes 3D-RoPE to model the relative positions of video tokens, where the pixel at the $t$-th frame and spatial location $(i, j)$ is assigned a RoPE index $(t, i, j)$. For the image tokens, we assign them to the $-1$-th frame (i.e., preceding the first video frame). Furthermore, to prevent the model from simply copy-pasting the image onto the video, we introduce a spatial shift as follows:

$$\text{Pos}(i, j) = \text{RoPE}(-1, i + w, j + h), \qquad (4)$$

where $w$ and $h$ denote the width and height of the video, respectively.

## 4 EXPERIMENT

### 4.1 EXPERIMENT SETTINGS

**Implementation Details.** The design of our model does not lead to task conflict issues. As shown in Table 1, we list the input conditions required for each task, where '1' indicates that the task requires the corresponding condition and '0' means it does not. We also add the corresponding text from the table to the beginning of the prompt. For example, in the Mask-guided video edit task, we refine the original prompt "A brown teddy bear is lying in the pot." to "Mask-guided. A brown teddy bear is lying in the pot." to further improve the model's ability to differentiate between different tasks. We adopted a phased strategy for the training process. We first train the mask-guided video edit task, which easily allows the model to extend to object movement, inpainting, and outpainting tasks. The introduction of mask information enables the model to effectively learn the ability to generate context

Figure 4: Qualitative of comparison on the wild dataset.

in the spatiotemporal dimensions. After extending the tasks to object movement, inpainting, and outpainting, the model has already learned a good correspondence between objects and text, which significantly reduces the difficulty of training the instruction edit task in the second phase. Finally, since the aforementioned edit tasks have already learned the correspondence between characters and masks, extending it to the controllable character video synthesis task only requires fitting the pose information. This training process greatly reduces the time cost of training individual tasks and enhances the model's performance.

**Datasets.** To obtain the high-quality data required, we used PySceneDetect PySceneDetect (2025) to segment transition videos into single-shot videos, Textbpn-Plus-Plus Zhang et al. (2023b) to filter out videos with excessive subtitles, and the Koala-36M Wang et al. (2024a) model to further refine our data selection. To extract the objects in the videos, we first used the Qwen-7B Bai et al. (2023) model to extract all object IDs in the videos. For portrait data, we used ArcFace Deng et al. (2019) to locate faces to ensure detection accuracy and filtered out the IDs that appeared in the most frames. Based on keywords, we used Grounding Sam2 Ren et al. (2024) to extract object masks and bounding boxes, discarding objects that were excessively large or small. Due to size differences between objects, we randomly expanded the masks in all four directions to mitigate the issue of overly restrictive masks. The Table 1 shows the data distribution used for each task.

Table 1: Input modalities and data distribution for each video editing task.

| Task | Image | Pose | Mask | Masked | Prompt | ‖ Human | No-Human | Total |
|---|---|---|---|---|---|---|---|---|
| Controllable character | 1 | 1 | 1 | 1 | Controllable | 294,822 | 0 | 294,822 |
| Mask-guided | 1 | 0 | 1 | 1 | Mask-guided | 102,374 | 198,653 | 301,027 |
| Human Animation | 1 | 0 | 0 | 0 | Human Animation | 293,567 | 0 | 293,567 |
| Inpainting | 0 | 0 | 1 | 1 | Inpainting | 148,731 | 151,089 | 299,820 |
| Outpainting | 0 | 0 | 1 | 1 | Outpainting | 199,405 | 101,678 | 301,083 |
| Object Addition | 1 | 0 | 0 | 0 | Object Addition | 148,731 | 151,089 | 299,820 |

Furthermore, due to the absence of a publicly available unified multi-task dataset, we have developed the OmniV2V-Test dataset. This test set comprises 100 pairs for each task, encompassing a variety of species, styles, and more. The diverse data distribution within the testset is designed to thoroughly evaluate the capabilities of various models.

**Evaluation Metrics.** To evaluate the model performance, we use the following metrics to measure the object consistency in videos, text-video alignment, and video generation quality: **ID consistency(Face-sim).** We employ RetinaFace Deng et al. (2020) and Arcface (Deng et al., 2019) to detect and extract the embedding of the reference face and each frames of generation video, and then compute the average cosine similarity between them. **Object similarity(DINO-sim).** First, we detect each frame and get the segment result of human using YOLOv11 (Khanam & Hussain, 2024),

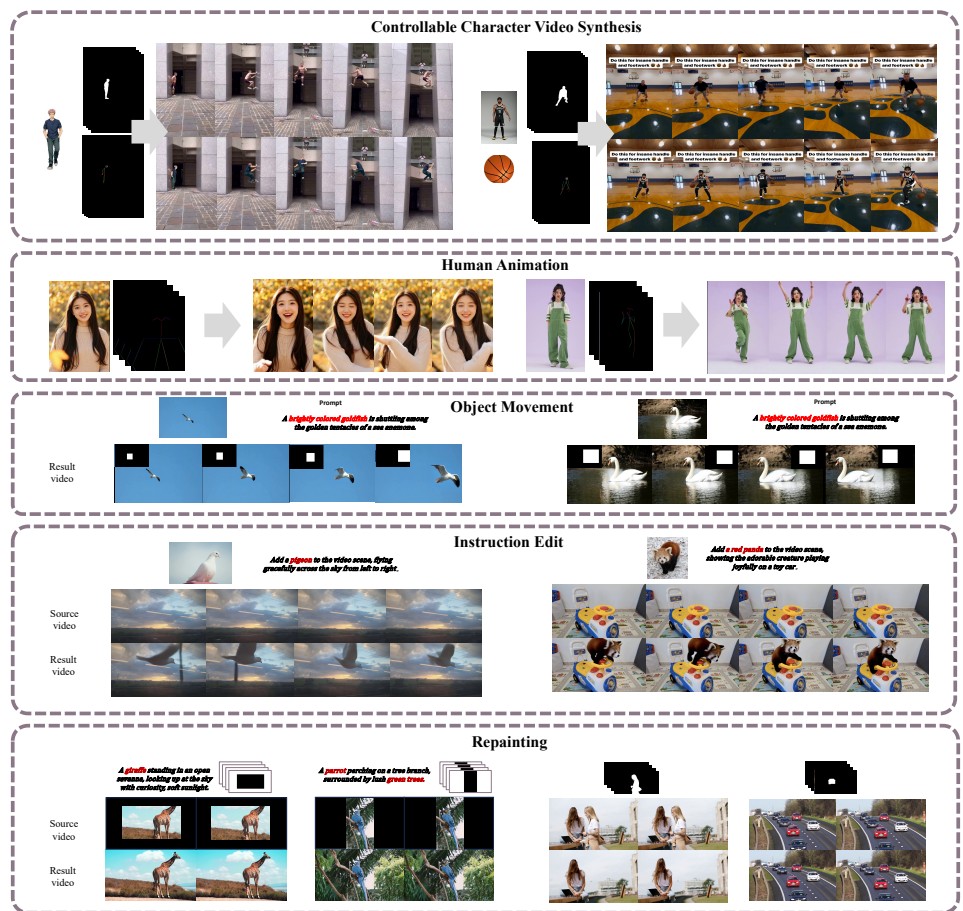

Figure 5: Visualization of videos generated by OmniV2V on the wild dataset.

and then compute the similarity of the DINO-v2 (Oquab et al., 2023) feature between the reference and results. **Text-video alignment.** We employ CLIP-B,CLIP-L (Radford et al., 2021) to evaluate the alignment between the given text prompt and the corresponding generated videos. **Temporal consistency(Temporal).** Following VBench (Huang et al., 2024), we utilize the CLIP-B (Radford et al., 2021) model to calculate the similarity between each frame and its adjacent frames, as well as the first frame, to assess the temporal consistency of the video. **Dynamic degree(DD).** The dynamic degree is used to measure the movement of an object, which is calculated following VBench.

**Compared Baselines.** We compare with specialized models for each task. For some tasks lacking open-source methods, we use commercially available online models as substitutes. The specific tasks can be divided into: (1) Repainting tasks, where in inpainting we mainly compare ProPainter Zhou et al. (2023) , VACE 14B Jiang et al. (2025) and VideoPainter Bian et al. (2025), and in outpainting, we mainly compare M3DDM Fan et al. (2023) and the VACE 14B; (2) mask-guided video edit tasks, where we mainly compare Kling 1.6 Keling (2025) and VACE 14B Jiang et al. (2025); (3) Instruction edit tasks, for which there are no corresponding open-source models, we mainly compare the commercial products Kling 1.6 and Pika Pika (2025); (4) Character Animation tasks, where we mainly compare Animate Anyone Hu (2024), Mimicmotion Zhang et al. (2024), and Champ Zhu et al. (2024), UniAnimate Wang et al. (2024b) and WanAnimate Wang et al. (2025), and for Controllable Character Video Synthesis tasks, we compare Mimo Men et al. (2024) and WanAnimate.

## 4.2 MAIN RESULTS

**Qualitative Results.** As shown in the figure 4, we conducted comparisons with existing approaches on tasks such as mask-guided editing and try-on. Our method achieves more realistic and temporally consistent results on the wild dataset compared to VACE 14B and Kling 1.6, with objects in the edited regions exhibiting more natural motion. Of course, qualitative comparisons for other tasks can be

Table 2: Quantitative comparisons with mask-guided video edit baselines and controllable character video synthesis baselines.

| Method | Face-sim ↑ | DINO-sim ↑ | CLIP-B ↑ | CLIP-L ↑ | FVD ↓ | Temporal ↑ | DD↑ | SC ↑ | MD ↑ | VQ ↑ |
|---|---|---|---|---|---|---|---|---|---|---|
| VACE 14B | 0.587 | 0.576 | 0.330 | 0.274 | 1171.42 | 0.966 | 0.524 | 5.66 | 4.32 | 6.66 |
| Kling 1.6 | 0.343 | 0.582 | **0.346** | **0.276** | 1049.70 | 0.933 | 0.642 | 3.33 | 8.64 | 7.32 |
| OmniV2V-Unified | 0.614 | 0.591 | 0.328 | 0.274 | 900.35 | 0.967 | 0.696 | 8.67 | 8.77 | **8.56** |
| OmniV2V-Mask | **0.638** | **0.625** | 0.342 | 0.281 | 940.22 | 0.968 | **0.718** | **9.33** | **9.12** | 8.33 |
| Kling1.6 | — | 0.543 | 0.308 | 0.265 | 1055.88 | 0.812 | 0.546 | 7.50 | 2.88 | 3.38 |
| Pika | — | 0.588 | 0.313 | 0.268 | 997.45 | 0.837 | 0.662 | 6.67 | **5.77** | 6.99 |
| OmniV2V-Unified | — | **0.596** | 0.321 | **0.269** | 968.74 | 0.854 | **0.766** | 6.98 | 4.78 | 7.32 |
| OmniV2V-Addition | — | 0.590 | **0.328** | 0.274 | **900.35** | **0.967** | 0.699 | **8.32** | 3.33 | **7.50** |
| Mimicmotion | 0.603 | 0.437 | — | — | 1216.81 | 0.820 | 0.717 | 4.07 | 4.29 | 6.73 |
| Unimate-DiT | 0.612 | 0.451 | — | — | 1193.14 | 0.831 | 0.832 | 4.11 | 4.44 | 6.85 |
| Wan-Animate | 0.609 | 0.445 | — | — | 1202.37 | 0.825 | 0.828 | 3.92 | 4.33 | 6.69 |
| OmniV2V-Unified | 0.618 | 0.587 | — | — | 998.04 | 0.964 | 0.849 | 5.48 | **6.69** | **8.84** |
| OmniV2V-Animation | **0.631** | **0.597** | — | — | **888.46** | **0.972** | **0.882** | **5.73** | 5.81 | 6.99 |
| Kling1.6 | — | — | — | — | 1200.56 | 0.754 | 0.664 | 7.98 | 7.14 | **7.66** |
| VACE14B | — | — | — | — | 960.21 | 0.815 | **0.688** | 7.68 | 7.00 | 6.43 |
| OmniV2V-Unified | — | — | — | — | **942.38** | 0.856 | 0.669 | 8.23 | 7.65 | 7.55 |
| OmniV2V-Inpaint | — | — | — | — | 963.98 | **0.884** | 0.671 | **8.67** | **8.77** | 6.56 |
| VACE14B | — | — | 0.342 | **0.284** | 1122.56 | 0.804 | 0.556 | 6.98 | **9.64** | **8.94** |
| OmniV2V-Unified | — | — | 0.338 | 0.272 | 984.24 | **0.841** | 0.643 | 7.02 | 7.65 | 8.82 |
| OmniV2V-Outpaint | — | — | **0.346** | 0.282 | **803.21** | 0.831 | **0.747** | 8.22 | 8.77 | 8.56 |
| Mimo | 0.446 | **0.562** | — | — | 1088.56 | 0.802 | 0.553 | 3.33 | 5.00 | 7.66 |
| Wan-Animate | **0.616** | 0.451 | — | — | 1189.52 | 0.822 | 0.821 | 3.96 | 4.38 | 6.75 |
| OmniV2V-Unified | 0.593 | 0.553 | — | — | 862.21 | 0.856 | 0.646 | 8.23 | 7.65 | **8.55** |
| OmniV2V-Control | 0.613 | 0.558 | — | — | **842.33** | **0.869** | **0.687** | **8.38** | **8.53** | 8.12 |

Table 3: Comparison of animation models on TikTok dataset.

| Model | SSIM↑ | PSNR↑ | LPIPS↓ | FVD↓ |
|---|---|---|---|---|
| AA | 0.718 | 29.56 | 0.285 | 171.90 |
| Mimic | 0.795 | 20.10 | 0.212 | 150.23 |
| Champ | 0.802 | 29.91 | 0.234 | 160.82 |
| Uni | 0.811 | 30.77 | 0.231 | 148.06 |
| Uni-DiT | 0.813 | 30.01 | 0.229 | 145.22 |
| Wan-Ani | **0.823** | 31.92 | **0.209** | 148.22 |
| Ours | 0.821 | **32.43** | 0.218 | **142.38** |

Table 4: Comparison on YouTube-VOS and DAVIS datasets.

| Model | YouTube-VOS | | | | DAVIS | | | |
|---|---|---|---|---|---|---|---|---|
| | PSNR↑ | SSIM↑ | LPIPS↓ | FVD↓ | PSNR↑ | SSIM↑ | LPIPS↓ | FVD↓ |
| M3DDM | 20.20 | 0.7312 | 0.1854 | 66.62 | 20.26 | 0.7082 | 0.2026 | 300.00 |
| VACE14B | 23.44 | **0.8601** | 0.1662 | **50.44** | 26.97 | **0.8582** | 0.1720 | 269.66 |
| Ours | **24.21** | 0.8545 | **0.1643** | 55.78 | **30.22** | 0.8422 | **0.1650** | 250.71 |
| Propainter | 19.85 | 0.7261 | 0.2010 | 92.21 | 20.03 | 0.7342 | 0.1984 | 325.00 |
| VACE14B | 22.50 | 0.8410 | 0.1725 | 65.80 | 25.61 | 0.8371 | 0.1757 | 310.23 |
| VideoPainter | 21.34 | 0.7512 | 0.1892 | 88.33 | 21.90 | 0.7284 | 0.1935 | 305.70 |
| Ours | **24.90** | **0.8583** | **0.1602** | **52.31** | **31.11** | **0.8550** | **0.1600** | **248.56** |

found in the supplementary materials. Moreover, our model can effectively integrate conditions from different modalities. As illustrated in the figure 5, we showcase the performance of our model across various sub-tasks, demonstrating its strong potential in the field of video generation and editing. More visualization results are available in the supplementary.

**Quantitative Results.** To further comprehensively validate the superiority of our method, OmniV2V-Unified, across various tasks, we conducted extensive comparisons on OmniV2V-Test with a range of task-specific approaches. In addition, we compared OmniV2V-Unified with the OmniV2V-[task] models trained individually for each task. As shown in the Table 2, our model outperforms existing baselines across different tasks. Moreover, the comparison between OmniV2V-[task] and OmniV2V-Unified demonstrates that our method does cause some forgetting for certain tasks, but all within an acceptable range. This also demonstrates that the HunyuanVideo-13B model is fully capable of accommodating these tasks.

**User Study** To further validate the effectiveness of our proposed method, we conducted evaluations on the objective assessment dataset of the OmniV2V-Test benchmark. Each participant assessed three key dimensions: Subject Consistency (SC), Motion Dynamic(MD), and Video Quality(VQ). A total of 30 participants scored each aspect on a scale from 0 to 10. As shown in the Table 2, the results indicate that OmniV2V outperforms all existing baseline methods across all evaluated dimensions. Notably, it achieves particularly significant improvements in motion dynamic and object consistency. The evaluation clearly demonstrate the superiority of our approach.

## 4.3 Ablation Study And Discussion

Table 5: Ablation on condition injection methods

| Method | DINO-sim↑ | CLIP-B↑ | CLIP-L↑ | FVD↓ | Temporal↑ | DD↑ |
|---|---|---|---|---|---|---|
| ChannelCat + Fc | 0.51 | 0.309 | 0.260 | 1233.90 | 0.942 | 0.55 |
| TokenCat | **0.54** | **0.336** | 0.262 | **984.82** | 0.958 | **0.59** |
| Ours | 0.53 | 0.312 | **0.263** | 1045.88 | **0.959** | 0.58 |

Table 6: Ablation study on token fusion.

| Task | Method | DINO-sim↑ | FVD↓ | Temporal↑ | DD↑ |
|---|---|---|---|---|---|
| (1) | w/o FC | 0.544 | 1200.96 | 0.662 | 0.63 |
| (1) | w FC | **0.55** | **862.21** | **0.86** | **0.64** |
| (2) | w/o FC | 0.548 | 980.49 | 0.942 | 0.57 |
| (2) | w FC | **0.59** | **900.35** | **0.97** | **0.69** |

**Ablation on token fusion of FC.** The Table 6 demonstrates the effectiveness and necessity of the FC layer in the Token Fusion process for tasks such as (1) controllable character video synthesis, (2) mask-guided video editing. The FC layer effectively allows the model to selectively retain or discard features, ensuring that only the most salient conditional information is integrated.

**Ablation on token fusion of condition injection methods.** We have explored three different condition injection methods: Channel Concat + Fc, Token Concat, and Addition. As shown in the table 5, Token Concat achieves the best performance on the mask-guided video editing task. However, since Token Concat doubles the GPU memory usage and inference time, it is not an ideal approach. Therefore, we chose element-wise addition for condition unification.

**Ablation on VLLM-driven instruction-based editing module.** In the Addition experiment, we validated the effectiveness of our VLLM-driven instruction-based editing module. As shown in Figure 6(a), removing 3D-RoPE causes the model to simply copy the image into the video, indicating that the spatial shift we introduced is effective. Additionally, the instruction tokens and image prompt tokens significantly aid the model in understanding the video content and performing relevant edits.

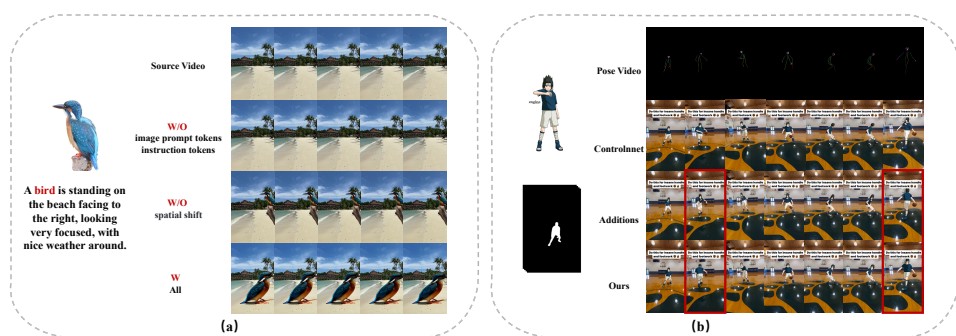

Figure 6: Ablation on VLLM-driven instruction-based editing module (a) and posenet(b).

**Ablation on posenet.** We evaluate how three different methods of injecting pose information affect the model's ability to learn pose information. This evaluation is conducted on the controllable character video synthesis task. All three experiments are tested after 3000 training steps. As shown in the figure 6(b), injecting pose information using the token addition method leads to the model failing to properly understand the front and back of the character, making it unable to capture actions such as turning around. The controlnet-based method results in slow or even incorrect learning of character movements. Our method effectively addresses the issues present in the aforementioned approaches.

## 5 Conclusion

In this paper, we explore a unified dynamic content manipulation injection module that effectively integrates the requirements of various tasks. To enhance the model's ability to understand the correspondence between visual content and text, we design a visual-text instruction module based on LLaVA. Given the numerous subtasks involved, we have developed a comprehensive multi-task data processing system. Since there is data overlap among various tasks, this system efficiently provides data augmentation. Using this system, we have constructed a multi-type, multi-scenario OmniV2V dataset, which significantly enhances the model's capabilities. Additionally, we have developed the corresponding OmniV2V-Test benchmark. The extensive distribution of test data allows for a thorough evaluation of model performance across various tasks. Both qualitative and quantitative experiments demonstrate that OmniV2V shows significant improvements over the best current open-source and commercial models in various video generation and editing tasks.

## STATEMENT

**Reproducibility statement**    We have explained the implementation of OmniV2V in detail in Sec. 3 and Sec. A.5. The code and dataset pipeline used in this work will be open-source online.

**Ethics statement**    As intelligent video generation and editing technologies become increasingly widespread, society is also facing new challenges. The convenience of generating and editing video content may lead to the spread of misinformation and false content, undermining public trust in information. At the same time, these technologies may inadvertently reinforce existing biases and stereotypes during content creation, negatively impacting cultural perceptions within society. These issues have sparked deep reflection on ethics and responsibility, prompting policymakers, technology developers, and all sectors of society to work together to establish appropriate regulations to ensure the healthy development of these technologies. We should also approach their potential impacts with caution, actively seeking a balance between innovation and social responsibility so that these technologies can bring greater benefits to society.

This technology has the potential to be used for generating misleading or deceptive videos, which could contribute to the spread of disinformation and fraudulent content, or even be exploited to manipulate public opinion or create social panic, resulting in financial losses for victims and harm to democratic institutions. However, our research is intended to promote the positive applications of this technology in creative and entertainment fields, rather than for impersonating real individuals. We strictly prohibit the unauthorized use of others' likenesses or artistic works, as this could infringe upon portrait rights and copyrights, leading to legal and ethical risks. We are fully aware of our responsibilities in the process of technological development; therefore, we always focus on enhancing the authenticity and quality of video content and firmly oppose any form of impersonation or fabrication of video content.

To minimize the potential negative impacts of this technology, we recommend the implementation of technical measures, such as video watermarking, content traceability, and automatic filtering—in commercial applications, to enhance the traceability and security of content. We are committed to maintaining full transparency regarding the capabilities and limitations of OmniV2V, and will strive to address potential bias issues as we continue to improve the model. We believe that the technological advancements brought by OmniV2V will help promote the positive application of AIGC in film production, assistive services, educational content, and other fields, improving industry efficiency, reducing repetitive labor, shortening production cycles, and accelerating the development of related industries.

At the same time, we encourage the research community to continue developing and refining synthetic content detection technologies while improving the quality of video generation. We believe that responsible innovation and proactive risk management are essential to ensuring that such technologies benefit society and are not misused.

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

# A APPENDIX

## A.1 THE USE OF LARGE LANGUAGE MODELS

We acknowledge that a large language model (LLM) was utilized solely for language editing and grammatical improvements during the preparation of this manuscript. The LLM was not involved in any key aspects of the research, including conceptualization, experimental design, data analysis, or interpretation of results.

## A.2 MORE VISUALIZATION RESULTS

As shown in the figure 8 and figure 9, we present more visual results for the controllable character video synthesis task and mask-guided edit task. It can be seen that our method demonstrates strong generalization across various scenarios. Moreover, our model largely addresses the issue of object shape mismatch caused by mask boundaries.

As shown in the figure 11, we present more visual results for both inpainting and outpainting tasks. our model effectively identifies small and fast-moving objects and successfully removes them, demonstrating the model's ability to handle complex scenarios with precision and efficiency. For the outpainting task, our model demonstrates the ability to generalize across various styles, such as anime and traditional chinese painting. This versatility highlights the model's adaptability and effectiveness in handling diverse artistic expressions.

As shown in figure 12 and figure 10, we provide additional visual results for the human animation and instruction edit tasks. In the human animation task, our method can accurately drive characters in various styles based on pose information, fully demonstrating the exceptional generalization capabilities of our approach. In the task of instruction edit, our method demonstrates impressive capabilities by directly replacing a bus in the video with a fire truck based on the given instructions, without the need for masking. This highlights the efficiency and precision of our approach in seamlessly handling complex video editing. Additionally, we showcase a scene where a woman is explaining cosmetics, illustrating the potential application of our model in the live streaming domain. By leveraging the capabilities of the model, users can easily modify visual elements to suit various backgrounds and themes, thereby expanding the creative horizons of digital media production.

## A.3 MORE EXPERIMENTS RESULTS.

**More Qualitative Results.** As shown in the figure 7, our model achieves better results compared to both open-source and commercial methods in other tasks. Specifically, in the instruction addition task, our method is able to understand the information in the text while reducing the problem of the model faithfully replicating the original image. In the inpainting task, we found that the Kling1.6 Keling (2025) model always tries to modify content outside the mask, resulting in lower video quality. In the outpainting task, VACE14B Jiang et al. (2025) fails to generate boundary extensions that match the textual descriptions well. In the controllable character video synthesis task, we mainly compare with the open-source model Mimo Men et al. (2024) and Wan-Animat Wang et al. (2025). It can be seen that our method achieves better subject similarity than Mimo and Wan-Animate.

## A.4 PRELIMINARY

In the training process, we adopt the Flow Matching (Lipman et al., 2022) framework to train the video generation models. For training, we first acquire the video latent representation $z_1$ and the corresponding identity image $I$. Then, we sample $t \in [0, 1]$ from a logit-normal distribution (Esser et al., 2024) and initialize the noise $z_0 \sim N(0, I)$ according to the Gaussian distribution. After that, we construct the training sample $z_t$ through linear interpolation. The model aims to predict the velocity $u_t = \frac{dz_t}{dt}$ conditioned on the target image $I$, which is used to guide the sample $z_t$ towards $z_1$. The model parameters are optimized by minimizing the mean-squared error between the predicted velocity $v_t$ and the real velocity $u_t$, and the loss function is defined as:

$$\mathcal{L}_{generation} = \mathbb{E}_{t,x_0,x_1} \left\| v_t - u_t \right\|^2 . \tag{5}$$

To endow our model with a more extensive representational capacity and enable it to capture and learn a broader range of complex patterns, we fully fine-tune the weights of both the pretrained video generation model and the LLaVA model, ultimately unlocking its full potential for delivering superior video customization results.

## A.5   IMPLEMENTATION DETAILS

During training, we initialize the model with the weights of HunyuanVideo13B Kong et al. (2024b), and keep the parameters of LLaVA and 3DVAE frozen, updating only all other parameters. The training is divided into two stages: in the first stage, we use 128 GPUs (each with 96GB memory), set the training video resolution to 540×896, the global batch size to 64, and the learning rate to 1e-5, training for 10,000 steps; in the second stage, we use 256 GPUs (each with 96GB memory), set the training video resolution to 720×1280, the global batch size to 128, and the learning rate to 3e-5, training for 20,000 steps.

## A.6   DATASET

**Details.**   For different tasks, we need to perform customized data operations: (1) For the inpainting task, we randomly inpaint the video based on different object masks. For the outpainting task, we randomly crop the original video and use the bounding box of the crop as our conditional input. (2) For the addition task in the instruction edit task, we can effectively use the before-and-after data from the inpainting task as pairs. For the swap task in the instruction edit task, we can effectively use the trained mask-guided video edit model to generate pairs. (3) For the Character Animation task and the controllable character video synthesis task, we used DWpose Yang et al. (2023) to extract the actions of characters in the videos. Due to significant differences in body types between characters, we also performed body type data augmentation on the DWpose data.

**Data augmentation.**

- **Enhancing Data Diversity:** Taking mask-guided video editing as an example, our data primarily involves two categories: human and non-human. However, many videos do not contain only a single category—instead, they often include multiple objects simultaneously. For instance, in the video description "a girl in a yellow dress playing the piano, with a yellow cup placed beside her," there are three distinct objects: "girl," "piano," and "cup." By extracting and labeling each of these objects separately from the same video, we can generate multiple training samples from a single clip. This approach not only expands the dataset effectively but also significantly enhances the diversity and richness of the training instances.

- **Increasing Data Volume:** During the data preprocessing stage, we employ a scene transition detection algorithm to split long videos into shorter segments. From these segmented clips, we randomly sample different video snippets as individual training samples. This strategy substantially increases the overall amount of available data.

- **Caption Augmentation:** For each video, we generate task-specific captions to support different editing objectives. For the Object Addition task, the prompt is formulated as: "Add a hat on the girl's head." In contrast, for the mask-guided video editing task, the caption emphasizes the editable region with precise localization, such as: "a girl wearing a yellow dress." This targeted captioning helps the model better understand and focus on the specific region to be edited, improving both accuracy and controllability.

## A.7   LIMITATIONS AND SOCIETAL IMPACTS

**Limitations**   Our method demonstrates strong capabilities across various video-to-video tasks. However, since the instruction edit task is primarily driven by text input, this signal is relatively weaker compared to signals such as mask or pose. As a result, there are scenarios where the instruction signal is ineffective. For example, when replacing very small objects (such as insects), the model may fail to accurately identify and replace the target object.

Meanwhile, in the controllable character video synthesis task, we observe that issues related to character interaction caused by mask boundaries still occur frequently. For instance, when replacing

a person holding a spear with a person holding a golden staff, the model often fails to generate the hands properly, resulting in poor interaction between the character and the object.

**Societal impacts** On the positive side, intelligent video generation and editing provide creators with a wealth of innovative tools, inspiring new ideas and enhancing the artistic and creative quality of video content. These technologies are gradually permeating various industries. For example, in the business sector, video generation technology is revolutionizing marketing and advertising strategies. Companies can quickly produce high-quality promotional videos, effectively communicate brand messages, and attract more consumers. This increase in efficiency not only reduces labor costs but also enables businesses to implement more creative marketing campaigns, thereby strengthening their competitiveness in the market.

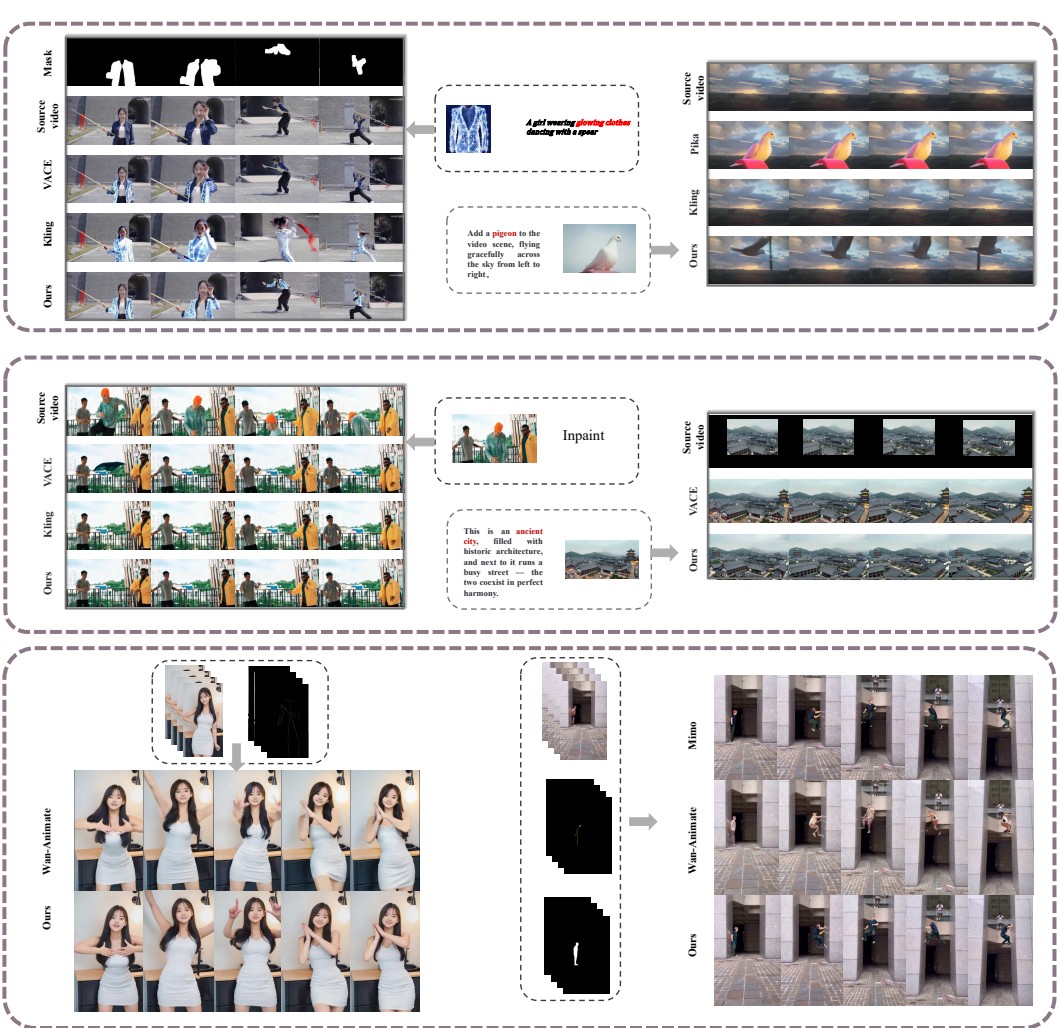

Figure 7: Qualitative comparison on all tasks.

A gray British Shorthair cat with round eyes stands on the ground paved with reddish-brown bricks.

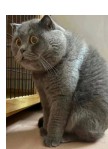
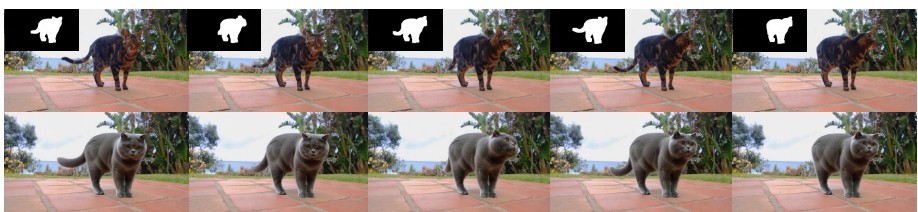

A girl is riding a white tiger running across the meadow.

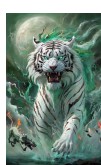
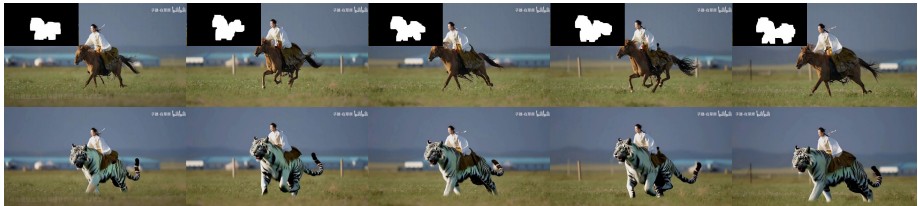

A joyful brown Corgi bounds towards the camera with enthusiasm from afar.

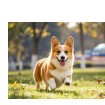
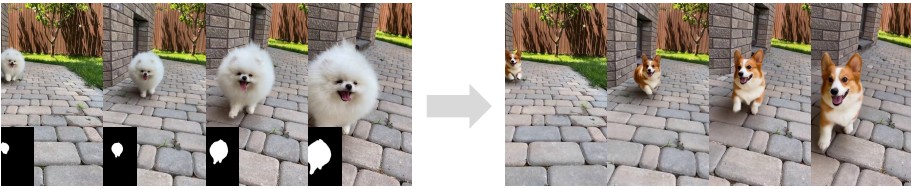

A brown dog is wearing a pair of red heart-shaped glasses.

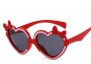
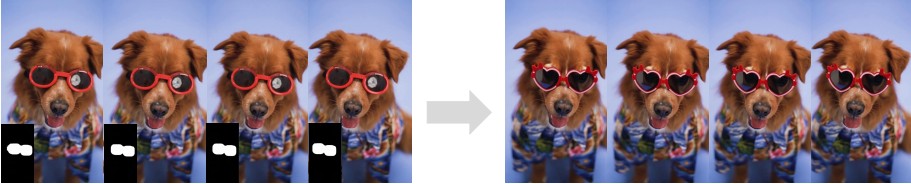

A blue jay (Cyanocitta cristata) is eating the food in a person's hand and then flies away.

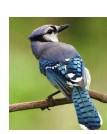
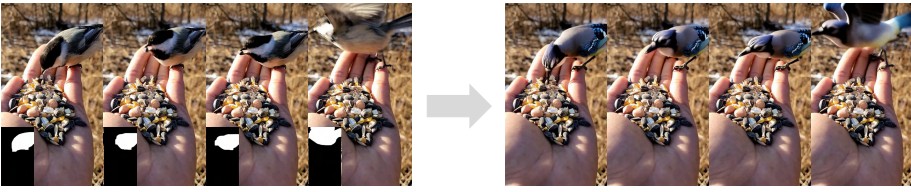

A  is playing a red guitar with a sharp body shape, and there are snow-capped mountains in the background.

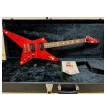
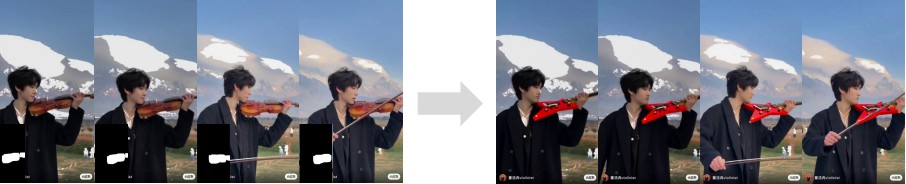

Figure 8: More visualizations of mask-guided edit task.

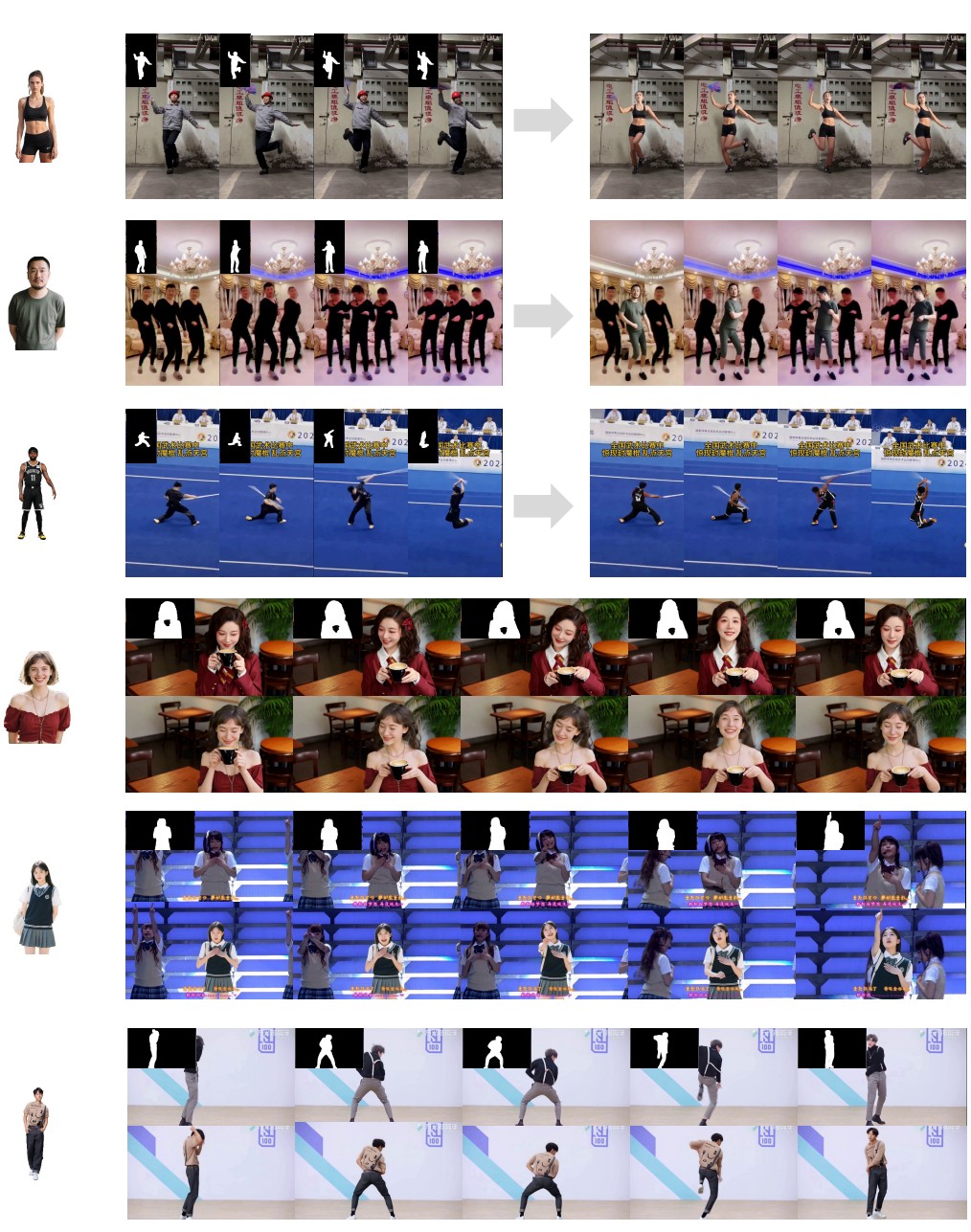

Figure 9: More visualizations of controllable character video synthesis task.

**Instruction Edit**

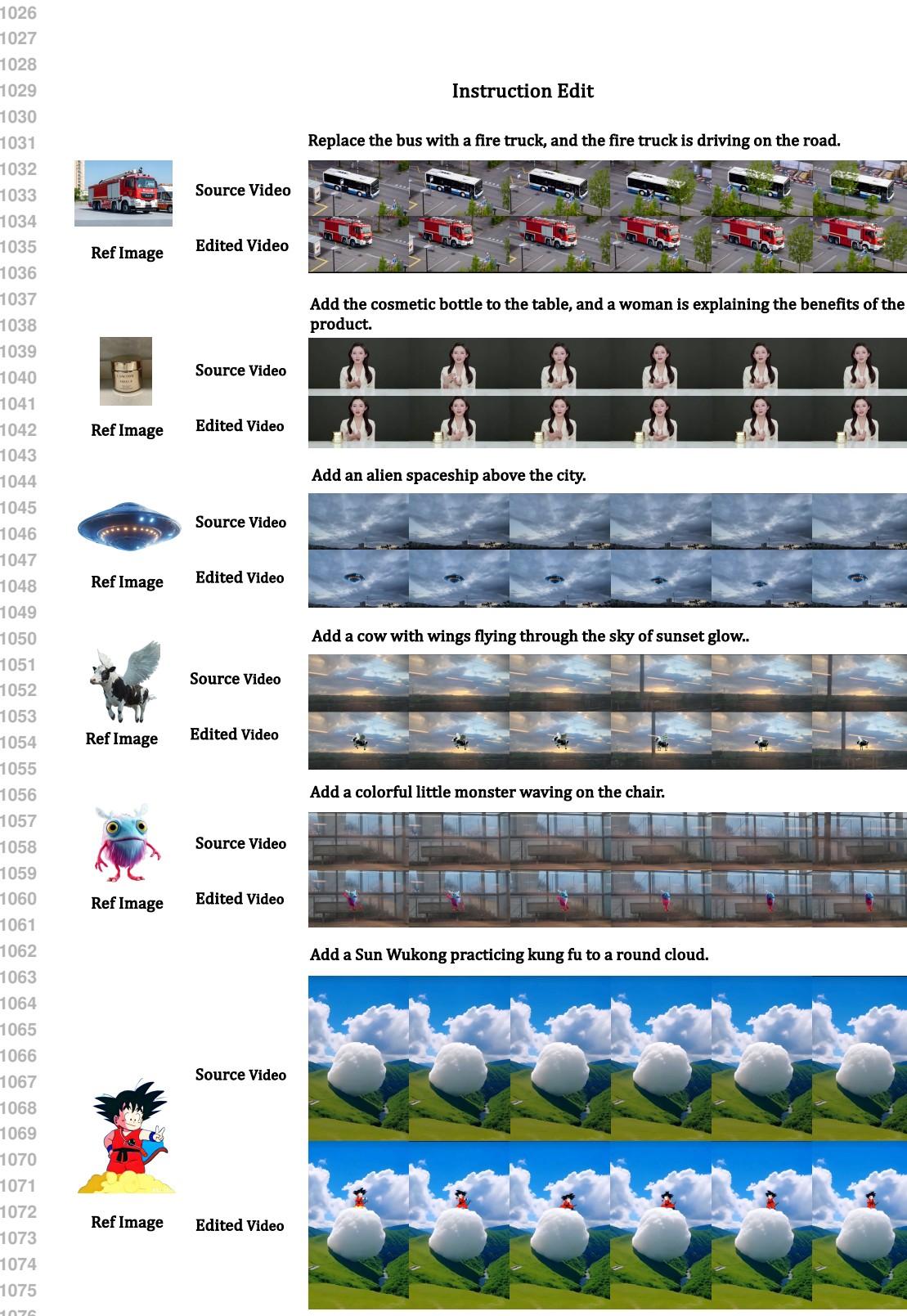

Figure 10: More visualizations of instruction edit.

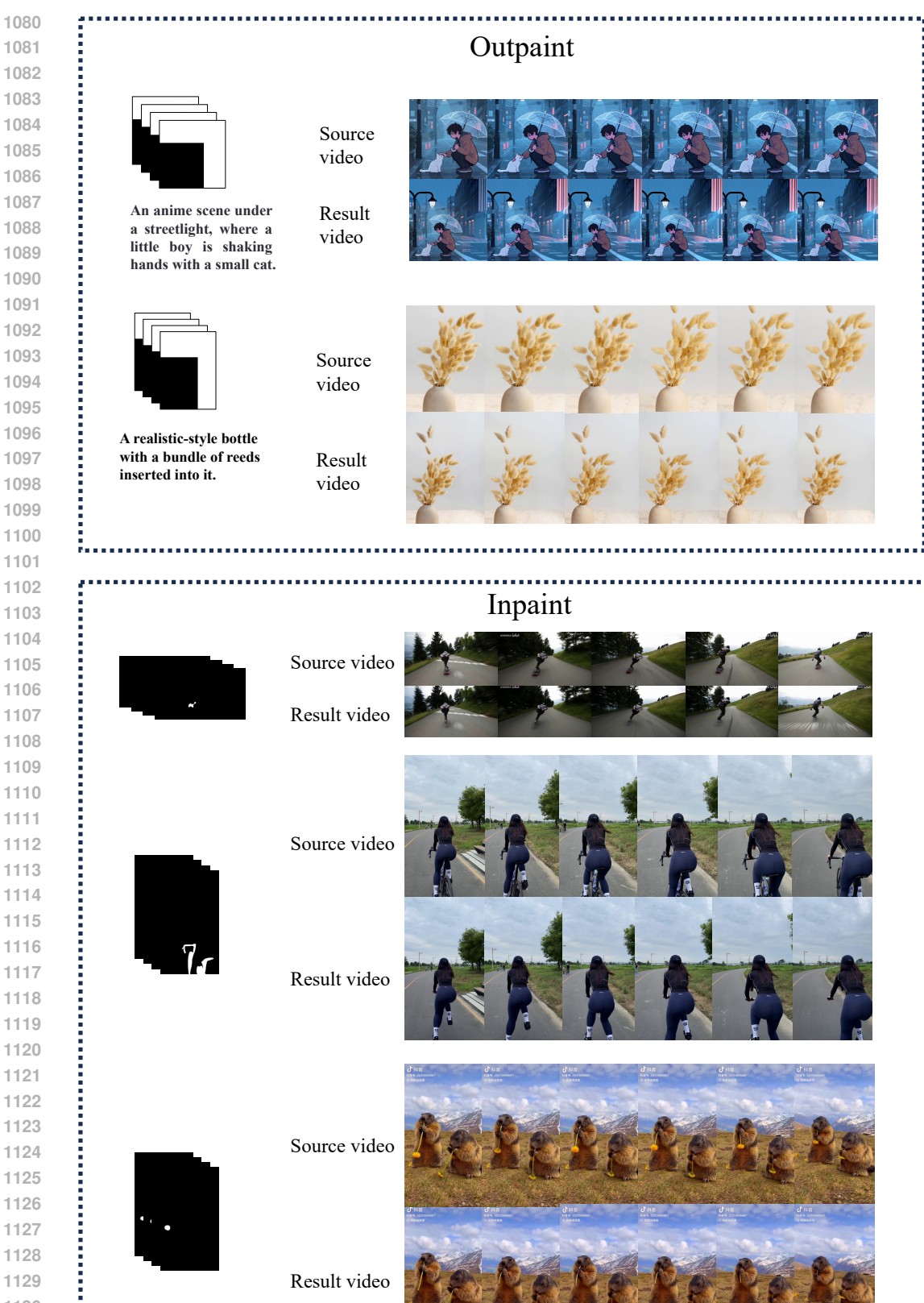

Figure 11: More visualizations of outpainting and inpainting task.

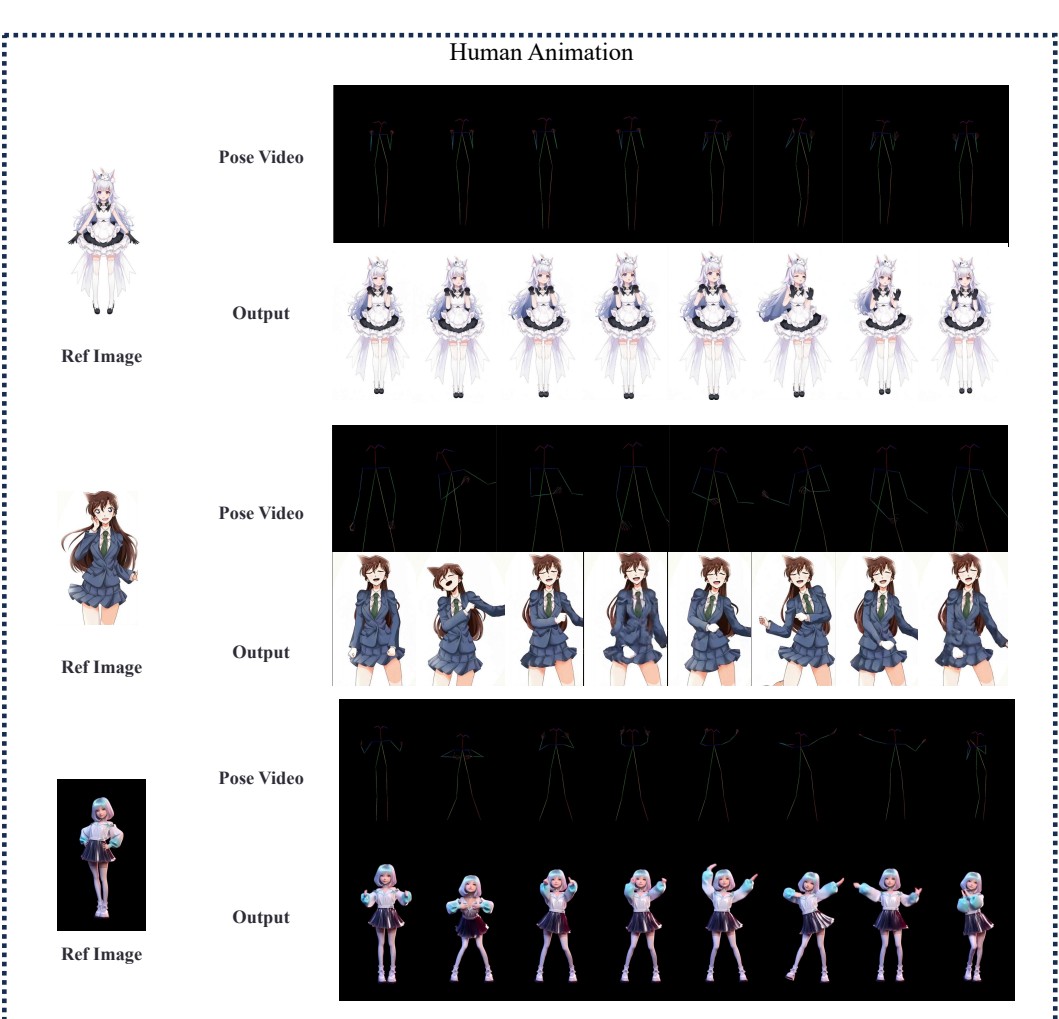

Figure 12: More visualizations of human animation and instruction edit task.

