# OpenReview forum: "OmniV2V: Versatile Video Generation and Editing via Dynamic Content Manipulation"
_ICLR.cc/2026/Conference — ICLR 2026 Conference Withdrawn Submission_

### Official Review · Reviewer_iugw · 2025-10-27

**Soundness:** 2
**Presentation:** 1
**Contribution:** 2
**Rating:** 2
**Confidence:** 5

**Summary:**

OmniV2V proposed a unified framework for both video generation and editing, capable of handling various tasks such as object addition and removal, inpainting and outpainting, and controllable character video synthesis etc. The approach features a dynamic content manipulation module that flexibly combines multi-modal inputs (images, videos, masks, poses) and a visual-text instruction alignment module built on LLaVA to more effectively translate user instructions into video edits. The authors also designed a multi-task data pipeline and contributed a evaluation benchmark. Experimental results show that OmniV2V performs competitively across diverse video generation and editing scenarios.

**Strengths:**

S1) It has a good motivation to supports a wide range of video generation and editing tasks within a single unified system.

S2) The content manipulation injection module and visual-text instruction module allowed efficient fusion of multi-modal inputs for flexible and adaptable content manipulation across diverse video generation and editing tasks. This contributes to the system’s versatility and usability for a wide range of applications within one unified framework.

S3) The design choice of dataset construction and evaluation metrics make sense to me, they are appropriate and well-justified.

**Weaknesses:**

W1) "We propose OmniV2V, a unified video generation and editing framework" The statement is misleading. "Video-to-video (V2V)" usually implies editing one input video to produce another, not general video synthesis/generation. As the short title only mentions video-to-video editing while the paper claims broader capabilities in both video generation and editing. The claim should be revised to make sure the title accurately reflect the full scope of the contributions.

W2) The method section is poorly written and unclear. In section 3.1, Naming conventions (e.g. Tokenizer1, Tokenizer3, v_md, v_mv) are used without explanations. The authors should rewrite for conciseness and introduce clearer variable naming, then provide step-by-step logic to improve readability.

W3) OmniV2V is limited to short-to-medium length video editing, as the duration and frame count limits inherent to its architecture.

W4) Missing prior V2V works references in related works and baseline comparison. E.g. CCEdit (CVPR 2024), AnyV2V (TMLR 2024), I2VEdit (SIGGRAPH Asia 2024), Ground-A-Video (ICLR 2024), IF-V2V (2025), VideoAnydoor (SIGGRAPH 2025) etc.

W5) The current in-text citations makes the paper less comfortable to read. Please revise it. E.g. LLaVA Liu et al. 2023 -> LLaVA (Liu et al., 2023)

Overall although the paper shows a good motivation and well-engineered design, the writing feels rough and must be heavily revised.

**Questions:**

Q1) Whats "ouwen" in Figure 2? Spelling issues?

Q2) The paper lists the construction of multi-task datasets and benchmarks as a contribution. Please clarify: will the datasets, benchmarks, and processing code be publicly released upon publication? Community dissemination would significantly enhance the impact and reproducibility of this work.

Q3) The maximum video length OmniV2V can generate is fundamentally limited by the underlying video generator architecture. However, in editing tasks, masked (source) videos might go beyond 20 seconds. It would greatly improve the clarity and practical relevance of the paper to include a discussion of these limitations and any strategies (current or future) for handling longer video sequences in editing scenarios.

---

### Official Review · Reviewer_ZtrF · 2025-10-29

**Soundness:** 3
**Presentation:** 1
**Contribution:** 3
**Rating:** 4
**Confidence:** 4

**Summary:**

This paper introduces OmniV2V, a method that wraps many video generation/edit tasks (mask-guided edits, object add/move, try-on, in/out-painting, human animation, controllable character synthesis) into one MM-DiT-based model via (1) a unified dynamic content manipulation injection that fuses tokens from source/mask/pose/noise and (2) a VLLM(LLaVA) instruction module that combines instruction/text/image as multimodal prompts to the model. It also introduces the OmniV2V-Test benchmark, showing competitive or better quality and temporal/ID consistency than task-specific systems.

**Strengths:**

1. The proposed unified video editing framework is important and can be beneficial to future works on video editing.
2. The visual results for the various video editing tasks are appealing.

**Weaknesses:**

1. The paper needs a major improvement in its writing quality. There are a lot of notation inconsistencies (e.g. FC and Fc used interchangeably for representing the fully connected layer); Some figures are referred to as "figure" and others are "Figure"; The citation formats in the paper are incorrect; When citing other models, some model names are also incorrect (e.g. Stableviton should be StableVITON; Grounding Sam 2 should be Grounded SAM 2; ArcFace is referred to as both ArcFace and Arcface in the paper). Please carefully revise the paper regarding these issues.
2. The VLLM-driven instruction-based editing part lacks necessary details. How is the LLaVA model used when constructing the multimodal input prompt? Is the final layer output representation from the LLaVA model used as a condition to the MM-DiT, or is it used in some other ways? Which version of LLaVA is used here? Is it the original LLaVA checkpoint based on CLIP and Vicuna, or is it some newer versions, such as LLaVA-1.5/1.6 or LLaVA-OneVision? What is the size of this LLaVA model? Is it the 7B version, or is it a larger model, such as 13B?
3. Following question 2, since the LLaVA model is frozen during the entire training process, has the author explored how the capacity of this MLLM model affects the editing performance? In other words, will we see improved model performances when switching to larger models and higher capacity models (e.g. Qwen2.5-VL [1], InternVL3 [2], etc)?
4. Figure 2 shows that the pose video is first processed by PoseNet and then further processed by the VAE encoder. Equation 1 indicates that the pose video is first encoded by VAE and then processed by PoseNet. Figure 3 shows that the pose video is directly processed by PoseNet and then fed into the model, without going through the VAE. There are three different illustrations of this pose conditioning branch. Which one is the method that is actually used in OmniV2V?
5. Can the author provide more details about the OmniV2V-Test benchmark, in terms of its data sources and data format (e.g. what input data/ground truth is accompanied with each video)?

[1] Bai, Shuai, et al. "Qwen2. 5-vl technical report." arXiv preprint arXiv:2502.13923 (2025).

[2] Zhu, Jinguo, et al. "Internvl3: Exploring advanced training and test-time recipes for open-source multimodal models." arXiv preprint arXiv:2504.10479 (2025).

**Questions:**

1. "To extract the objects in the videos, we first used the Qwen-7B Bai et al. (2023) model to extract all object IDs in the videos." I didn't fully understand this description. What is the input for the Qwen model? Is it the video caption, or is it actually a Qwen VL model (e.g. Qwen2.5-VL) such that the video frames are input into the model for object ID extraction?
2. Will OmniV2V-Test be released?

---

### Official Review · Reviewer_QkKY · 2025-10-30

**Soundness:** 2
**Presentation:** 3
**Contribution:** 2
**Rating:** 4
**Confidence:** 4

**Summary:**

This paper introduces OmniV2V, a unified framework for video generation and editing that handles multiple tasks including object movement, object addition, mask-guided editing, try-on, inpainting, outpainting, human animation, and controllable character synthesis. The main technical contributions are: (1) a unified dynamic content manipulation injection module that integrates various conditional inputs through token fusion, (2) a visual-text instruction module based on LLaVA for aligning visual content with textual instructions, and (3) a comprehensive multi-task dataset with corresponding benchmarks. The model is built on HunyuanVideo-13B and demonstrates competitive performance across various tasks.

**Strengths:**

1. Successfully unifying 8 different video editing tasks in a single framework is a notable engineering achievement with practical value
2. The paper provides thorough quantitative metrics (Face-sim, DINO-sim, CLIP, FVD, temporal consistency) and user studies across multiple tasks
3. The combination of text, image, pose, and mask conditions through a unified architecture is well-executed from an engineering perspective

**Weaknesses:**

1. Core contributions (token fusion, using LLaVA, dynamic training strategy) are straightforward extensions of existing techniques
2. The "unified dynamic content manipulation injection module" is essentially concatenating/adding tokens with random dropout
3. Did you retrain any open-source baselines (Mimicmotion, Champ, UniAnimate, etc.) on your OmniV2V dataset? If not, how can you claim
   your method is better when it's trained on different (potentially better) data?
4. Some claims are vague: "ensures that only the most salient conditional information is incorporated" (line 189) - how is salience determined?

**Questions:**

1. Unfair baseline comparison: The paper trains OmniV2V on ~1.8M curated samples but compares against baselines using their original pretrained weights (trained on different datasets). This makes it impossible to determine whether performance gains stem from the proposed method or simply better training data. Open-source baselines should have been retrained on the same OmniV2V dataset for valid comparison.

2. The spatial shift in Eq. 4 (adding w and h to RoPE positions) appears ad-hoc without principled justification. Why would shifting by the full video dimensions prevent copy-pasting?

3. The latent fusion tokenizer (Eq. 2) simply concatenates features and pads zeros - this may lose important relational information between mask and content

4. The paper claims to use "dynamic routing strategy" but never formally defines what this routing entails. Section 3.1 only mentions "randomly set some conditional inputs to empty" during training, which is standard practice, not a novel routing mechanism.

I am open to engaging in thorough discussions with the authors during the rebuttal period. Overall, I am not fundamentally opposed to the acceptance of this work; however, several critical concerns must be adequately addressed before I can recommend acceptance.

---

### Official Review · Reviewer_Rq5r · 2025-11-01

**Soundness:** 3
**Presentation:** 3
**Contribution:** 3
**Rating:** 4
**Confidence:** 4

**Summary:**

The authors propose OmniV2V, a single, unified framework for a very wide range of video generation and editing tasks, including object addition/movement, inpainting/outpainting, human animation, and controllable character synthesis. The primary motivation is to move away from the current "one model per task" paradigm, which is inefficient and costly to train and deploy.
A "unified dynamic content manipulation injection module" that processes and merges diverse conditional inputs (like reference images, pose videos, and masks) into their MM-DiT backbone (HunyuanVideo). This is trained with a dynamic routing strategy (i.e., randomly dropping out conditions) to handle various input combinations.
A "visual-text instruction module" built on LLaVA that allows the model to understand complex instructions that combine text ("add a cat") and visual references ("that looks like this image").
A new multi-task dataset (OmniV2V) and benchmark (OmniV2V-Test) specifically designed to train and evaluate such a multi-faceted model.
The authors present extensive experiments against many recent open-source and commercial models (Kling, VACE, Pika, etc.), showing that their unified model performs competitively or even superiorly on these individual tasks.

**Strengths:**

1. The paper is well-written and easy to follow.
2. The problems this paper is solving are important. It directly attacks the costly and inefficient "one model per task" paradigm, which is a significant bottleneck in the field. A successful unified model would be a major contribution.
3. The evaluation is comprehensive: The authors have done a massive amount of work to evaluate their model. Comparing against a long list of SOTA models (VACE) and closed-source commercial models (Kling, Pika) provides a strong, grounded set of results. The new OmniV2V-Test benchmark is a solid contribution for evaluating future work in this area.

**Weaknesses:**

1. Benefit of Unification is Unclear: The paper's central premise is that unification is better, but the results in Table 2 seem to contradict this. The OmniV2V-Unified model is almost always slightly worse than the authors' own single-task trained models (OmniV2V-Mask, OmniV2V-Animation, OmniV2V-Control, etc.). This suggests that the unification introduces negative interference between tasks, forcing a performance trade-off.  The main benefit seems to be parameter efficiency (one model vs. many), not superior performance. It looks like the tasks trained together do not benefit from each other.

2. The paper didn't study the generalization ability of a unified model: for example, generalization to unseen task composition, which is a crucial benefit of unified modelling. The dynamic training strategy teaches the model to handle known combinations of inputs. But can it handle unseen combinations? For example, can it perform "controllable character synthesis" (which uses pose, mask, and image) and "object addition" (which uses an instruction and another image) at the same time? This would be a powerful demonstration that the model has learned a truly general-purpose editing capability, rather than just being a multi-headed switch for different tasks.

3. The visual-text instruction module appears to be a straightforward concatenation of an instruction prompt and a target video caption. This module lacks significant technical novelty as claimed in the contribution.

**Questions:**

Please see W1,2. which needs further clarification.

---

### Note · Authors · 2025-11-12

**Comment:**

Withdrawal

**Withdrawal Confirmation:**

I have read and agree with the venue's withdrawal policy on behalf of myself and my co-authors.